

# GPs' motivation for teaching medical students in a rural area—development of the Motivation for Medical Education Questionnaire (MoME-Q)

Charles Christian Adarkwah[1,2,3], Annette Schwaffertz[4], Joachim Labenz[5], Annette Becker[2] and Oliver Hirsch[6]

[1] Department of Health Services Research and General Practice, Faculty of Life Sciences, University of Siegen, Siegen, Germany
[2] Department of General Practice and Family Medicine, University of Marburg, Marburg, Germany
[3] CAPHRI School for Public Health and Primary Care, Department of Health Services Research, Maastricht University, Maastricht, The Netherlands
[4] Medical School, University of Giessen, Giessen, Germany
[5] Department of Medicine, Diakonie Klinikum Siegen, Siegen, Germany
[6] FOM University of Applied Sciences, Siegen, Germany

Corresponding author
Charles Christian Adarkwah,
charles.adarkwah@uni-siegen.de

## ABSTRACT

**Background.** The establishment of a medical education program in the rural area of Siegen is planned to be the first step against a shortage of physicians in this region. General practitioners (GPs) will be extensively involved in this program as Family Medicine (Allgemeinmedizin) will become a core subject in the curriculum nationwide. Based on this situation we aim to figure out GPs motivation to participate in medical education. For this purpose, we had to construct and test a new questionnaire.

**Methods.** A survey was conducted among general practitioners (GPs) in the region of Siegen-Wittgenstein regarding their motivation to participate in medical education. For this purpose, the Motivation for Medical Education Questionnaire (MoME-Q), a 24-item questionnaire, was developed. Structural characteristics of GPs, the Maslach Burnout Inventory (MBI) and the Work Satisfaction Questionnaire (WSQ) were used for validation purposes.

**Results.** A representative number of GPs took part in the study (53.8%). Although the majority had no connection to a university (86%), 83% can imagine participating in the education of medical students. The items of the MoME-Q load on two factors (commitment and personal benefit). The confirmatory factor analysis shows a good model fit. Subscales of the MoME-Q were able to differentiate between physicians with and without authorization to train GP residents, between practices with and without a specialized practice nurse, and between physicians with and without previous experience in medical education. The MoME-Q subscale "commitment" correlated significantly with all three subscales of the MBI. Correlations were in the medium range around |.30|.

**Conclusion.** The MoME-Q seems to be an appropriate tool to assess motivation to participate in medical education of GPs. In our sample, a large number of GPs was motivated to participate in the education of medical students. Future studies with larger number of GPs should be carried out to validate and confirm our findings. Whether

the MoME-Q is also appropriate for other specialties should also be shown in further empirical studies.

## INTRODUCTION

In many rural regions, a shortage of physicians, especially general practitioners (GPs) is obvious and will dramatically increase in the near future (*Adarkwah et al., 2018*; *Broermann et al., 2018*). A smaller number of GPs will have to take care of a larger number of patients and catchment areas will increase. Furthermore, GPs in the German setting will be challenged by the Masterplan 2020. With the Masterplan Medical Education 2020, the importance of ''General Practice" will significantly increase. General Practice will become a major subject within the medical education curriculum (*Bundesministerium für Gesundheit, 2017*). On the one hand, every student will have to complete three months of ambulatory patient care within the sixth study year (practical year); on the other hand, General Practice will become a mandatory examination subject in the final oral examination (3rd part of the examination). This fact is of high relevance as a large number of GP practices for teaching and training will be necessary in order to comply with this demand. Those two facts, the demographical perspective as well as the necessity to participate in the education of medical students, create a challenge that needs to be mastered.

The district of Siegen-Wittgenstein represents a typical and representative rural region in Germany. Here, a shortage of doctors, especially general practitioners (GPs) is obvious and will dramatically increase in the near future. Some municipalities in Siegen-Wittgenstein are already listed at the Ministry of Health as municipalitites, where a critical shortage of GPs has already occurred (Erndtebrück, Kreuztal, Wilnsdorf or Burbach) (*Ministerium für Gesundheit, Emanzipation, Pflege und Alter des Landes Nordrhein-Westfalen, 2016*) or is most likely to occur within the near future (e.g., Bad Berleburg, Bad Laasphe) (*Ministerium für Gesundheit, Emanzipation, Pflege und Alter des Landes Nordrhein-Westfalen, 2017*). In other words, a significant number of the GPs have reached retirement age and the total number is too low to provide sufficient medical service for the aging population in the district.

Offering a medical education program in such a region could therefore be seen as a first useful step to diminish the shortage of doctors, especially GPs, in the region in order to preserve medical care for patients, for instance in the rural area.

In the near future, medical students will be educated and trained in Siegen, where a new medical campus will be established in cooperation with the University of Bonn Medical School. Starting with the term 2018/19 twenty-five medical students will be enrolled for the Bonn/Siegen program (*Universität Siegen, 2018*). Students will start their education in Bonn for three 3 years and will then continue and finish their studies in Siegen (study year 4 to 6). Prior research has demonstrated a so called ''Klebeeffekt" for the field of medical

education, which means that a lot of medical students stay in the greater area where they finished their studies in order to work in hospitals and practices after passing their final exams (*Buxel, 2009*; *Lenz et al., 2010*; *Töpfer, Silbermann & Maertins, 2011*; *Jacob, Kopp & Schulz, 2015*). As up to now no medical school campus is available in Siegen, only a few GPs are collaborating with other universities regarding medical education and research. Hence, only a limited number of GP sees medical students in their practices on a regular basis. Having motivated teachers does have an impact on students' performance. Studies have shown significant effects of teachers' characteristics on the achievement of students (*Wayne & Youngs, 2003*; *Zumwalt & Craig, 2005*). For instance, the motivation of teachers can enhance autonomous learning motivation in their students (*Roth et al., 2007*; *Radel et al., 2010*; *Kunter et al., 2013*) which in the end can have a positive effect on the overall academic performance (*Kusurkar et al., 2013*).

Until now, there was no instrument available for the assessment of teaching motivation in the ambulatory setting for physicians without teaching experience. Next to qualitative studies (*Thomson et al., 2014*; *Ingham et al., 2015*) to assess motivation, the Physician Teaching Motivation Questionnaire (PTMQ) was developed to measure motivation to teach in physicians already involved in medical education (*Dybowski & Harendza, 2015*). It was developed and validated in physicians from internal medicine and surgery and shows good statistical quality criteria.

Based on this situation we aim to figure out GPs motivation to participate in medical education in the district of Siegen-Wittgenstein. Furthermore, we aim to look at structural characteristics and the GP's motivation to participate in medical education. For this purpose, a new instrument has to be developed and examined.

## MATERIALS & METHODS

### Design and GP recruitment

We conducted a study in which all general practitioners in the district of Siegen-Wittgenstein ($n = 158$) were invited to take part. Contact details are freely available on the website of the Association of Statutory Health Insurance Physicians in the region of Westphalia (https://www.kvwl.de/earzt/). GPs were asked regarding their motivation to participate in medical education of students as well as their work satisfaction and burnout risk. In addition, they were asked in detail regarding their future prospects of working as a GP. In this paper, we focus on the GPs motivation to participate in medical education of students.

This survey (HaMEdSi: **Ha**usärzte (GPs) for **M**edical **ED**ucation in **Si**egen-Wittgenstein) was performed in general practices in the area of Siegen-Wittgenstein in Germany between October 2017 and January 2018. GPs were sent a written invitation with a detailed study description, informed consent and the study questionnaire. After 4 weeks, all GPs who did not respond received a telephone reminder by a member of the study team. An invitation to participate was also sent by email to all members of the local doctor's association, in which most of the GPs hold a membership. Furthermore, an informative meeting on the medical education perspective at the University of Siegen was held to which all GPs were
invited and $n = 45$ took part. In this meeting, GPs were also reminded and invited to take part in the study.

The study was performed in accordance with the Declaration of Helsinki and approved by the research ethics committee of the University of Marburg (Az.: Studie 127/17).

### Development of the Motivation for Medical Education Questionnaire (MoME-Q)

As there is no appropriate tool available to assess GPs motivation to take part in medical education of students, we developed a questionnaire based on the existing literature (*Thomson et al., 2014*; *Ingham et al., 2015*) as already mentioned above. Further items were developed and consented by expert panel meetings involving GPs as well as medical specialists experienced in the training of medical students. Afterwards a small pilot study ($n = 6$) was conducted among GPs experienced in the training of medical students ($n = 3$) as well as among GPs with less experience in teaching students ($n = 3$). All participants were invited to give a detailed written feedback and were also interviewed for further feedback on the questionnaire. The initial version of the questionnaire was slightly adjusted after the pilot study and as a result the Motivation for Medical Education Questionnaire (MoME-Q) developed. The MoME-Q is a 24item questionnaire with a four-point Likert scale with verbal descriptions "agree", "slightly agree", "slightly not agree", "do not agree". Based on the critical reading of the literature (*Thomson et al., 2014*; *Ingham et al., 2015*) and the expert panel meetings, we hypothesized that the instrument would have a four-factor structure with factors "conviction", "personal benefit", "personal resources", and "time management" with lower scores meaning more positive outcomes on the respective scales.

### Further instruments

We used the German version of the Maslach Burnout Inventory (MBI) to assess occupational burnout. The MBI comprises of 22 items to be scored on a 7-point-scale from "0-never" to "7-every day". It consists of three subscales, namely emotional exhaustion (nine items) which measures exhaustion at work, depersonalization (five items), which measures loss of empathy and emotional distance to others, and personal accomplishment (eight items) which measures competence and positive attitude towards work. The three-factor structure was confirmed (*Neubach & Schmidt, 2000*). Cronbach-$\alpha$ of the emotional exhaustion scale was .85, of the personal accomplishment subscale .71, and of the depersonalization subscale just .48. Other studies found higher internal consistencies for this subscale with Cronbach-alphas of .69 and .86, respectively (*Schwarzer, Schmitz & Tang, 2000*; *Gumz et al., 2013*). Convergent and discriminant validity of the MBI could be demonstrated.

The Work Satisfaction Questionnaire is comprised of 17 items to be scored on a 7-point-scale from "1-very dissatisfied" to "7-very satisfied". It has a five-factor structure with factors patient care (4 items, Cronbach-$\alpha = .76$), burden (4 items, $\alpha = .79$), income-prestige (3 items, $\alpha = .83$), personal rewards (3 items, $\alpha = .71$), professional relations (2 items, $\alpha = .66$). Furthermore, a global item asks for the satisfaction with the current job situation. This item correlates with the subscale scores form .39–.71 (*Bovier & Perneger, 2003*).

## Statistical analyses

There were up to three missing values on single items of the MoME-Q. They were replaced by the k nearest neighbor algorithm (kNN) (*Beretta & Santaniello, 2016*) using the R package VIM (*Kowarik & Templ, 2016*).

We conducted a confirmatory factor analysis with the R package lavaan (*Rosseel, 2012*). We used the robust Unweighted Least Squares Estimator (ULSMV), as this estimation method makes no distributional assumptions (*Rosseel, 2012*; *Lei & Wu, 2015*). Different model-fit statistics were calculated. The $\chi^2$/df ratio is a badness-of-fit-index as smaller values indicate a better fit (*West, Taylor & Wu, 2015*). Values around 2 signal a good model fit. The Root Mean Square Error of Approximation (RMSEA) is a population-based index that relies on the noncentral $\chi^2$ distribution. It can be regarded as an "error of approximation" index because it assesses the extent to which a model fits reasonably well in the population (*Brown, 2015*). Values $\leq .08$ are considered to indicate an adequate model fit (*Browne & Cudeck, 1993*). The standardized root mean square residual (SRMR) was calculated that measures the mean absolute value of covariance residuals (*Little & Kline, 2016*). Values below .10 indicate a good model-fit (*Weiber & Mühlhaus, 2014*). The Comparative Fit Index (CFI) and the Tucker Lewis Index (TLI) were not considered as they are sensitive to smaller sample sizes like ours in ULS estimation (*Lei & Wu, 2015*). The resulting items and scales were examined by parameters based on classical test theory like Cronbach- $\alpha$, discriminatory power, average intercorrelations. Omega coefficients for the applied scales were also computed using R packages psych and GPArotation as they have known advantages over Cronbach's- $\alpha$ (*Raykov, 2001*).

We used Hotelling's $T^2$-test from the R library "Hotelling" to compare different demographic groups on the scales of the MoME-Q (*Hair, 2010*). After a significant multivariate result univariate Welch t-tests were calculated to explore the analyses further. The univariate effect size partial Cohen's d was then calculated with values of .20 representing a small effect, .50 showing a medium effect, and .80 a large effect (*Grissom & Kim, 2012*).

We used the Spearman correlation coefficient to calculate associations between the MoME-Q subscales and other instruments as most of the variables deviated significantly from the normal distribution (*Kim, Kim & Ergün, 2015*). Due to multiple testing, the significance value was adjusted by the Bonferroni correction (*Bortz & Schuster, 2010*).

# RESULTS

## Characterization of the study sample

The total population consists of 158 GPs. Of them, 85 (53.8%) took part in the study and completed the questionnaire. There are 64 male GPs (75.3%) in our sample. The gender distribution conforms to the proportions in the population in this specific area. Mean age of the participants is 53.5 years (SD 8.93) with a median of 54 years, a minimum age of 32 and a maximum age of 73 years. The majority (91.8%) are practice owners, work full-time (90.6%) and work in a group practice (67.1%). The average study participant works in private practice for 18.41 years (mean, SD 9.8 yrs.) with a range between 2 and

43 years. Most of them are specialized in General Practice (51.8%), whereas 24.7% are specialized in Internal Medicine and 20.0% have both specializations. The minority (3.5%) are "Praktischer Arzt" without any further specialization. This denomination has been disestablished and taken out the regulation for further education in 1992. It is notable that despite of all political obstacles 94% would become GPs again.

Looking at teaching, only 14% have an affiliation with a university. The majority (59%) has at least some teaching experience, for instance most GPs have seen students for a clinical elective (57%), whereas only 17% have seen students within a university primary care rotation program (Blockpraktikum). A minority of 11% has seen students for parts of the practical year and only 3% have ever given lectures or seminars at a medical school.

Table 1 summarizes the demographic characteristics of the study participants.

## Motivation for Medical Education Questionnaire (MoME-Q)

The original version of the Motivation for Medical Education Questionnaire consisted of 28 items Table S1). After inspection of the statistical characteristics of the items and supported by the explorative nature of the study, we decided to exclude 4 of them from further analyses (Table 2). Further details are explained below.

The item "I hope to attract more patients being an "Academic Teaching Practice"" was initially considered to have a positive connation. The study results show, that GPs in the region of Siegen-Wittgenstein, threatened by a critical shortage, do not desire to treat more patients. Consequently, 76.5% of responses are in categories "3" and "4" which means that physicians do not wish to attract more patients for their practices as their current workload obviously is already high. The item is highly left skewed ($p < .001$).

The item "Students can support and relieve me in daily routine patient care" can be interpreted differently. On the one hand, the GP as a teacher should focus on teaching and support a good learning environment and not make use of the student's work force in the first instance. On the other hand, integrating students in real patient care according to their state of knowledge can make sense and foster personal development. Furthermore, teaching students next to patient care in daily practice is also time-consuming. As the content of the item remained unclear, we decided to exclude it. The item "I hope that General Practice gets more attention if more GPs take part in medical education of students" is too general. Therefore, 76.5% of responses are in categories "1" and "2" which means that physicians hope that their specialty will get more recognition by participating in the education of students.

The item "I have made bad experiences with medical students in my practice in the past" has a mean of 3.81, a median of 4, and a SD of 0.15; 100% of responses are in categories "3" and "4" which means that physicians hardly ever made bad experiences with students. Most of the study participants did not make experiences with medical students at all, which made it impossible for the majority of study participants to answer this question.

Descriptive statistics of the remaining 24 items are displayed in Table 3. The numbering of the items in the article corresponds to the initial version of the questionnaire. As can be seen, a substantial number of items shows significant deviations from normality regarding skewness and kurtosis.

**Table 1  Demographic characteristics of study participants ($n = 85$).**

**General characteristics**

| | |
|---|---|
| Age | Mean 53.5 years |
| | SD 8.9 |
| Years in practice | Mean 18.4 years |
| | SD 9.8 |
| Gender | 64 male (75%) |
| | 21 female (25%) |
| Practice ownership | 78 practice owners (92%) |
| | 7 practice employees (8%) |
| Specialization | 44 General Practice (52%) |
| | 21 Internal Medicine (25%) |
| | 17 General Practice and Internal Medicine (20%) |
| | 3 none (3%) |
| Practice form | 28 single practice (33%) |
| | 57 group practice (67%) |
| Modus of work | 78 full-time (92%) |
| | 7 part-time (8%) |
| Would become GP again | 80 (94%) |

**Teaching experience**

| | |
|---|---|
| Cooperation with medical school/status of an academic teaching practice | 12 (14%) |
| Any preexisting teaching experience | 50 (59%) |
| One-day observation | 38 (45%) |
| Two-week rotation | 14 (17%) |
| clinical elective | 48 (57%) |
| Practical Year | 9 (11%) |
| Lectures at a university | 3 (3%) |
| Visited didactics training within last 2 years | 5 (6%) |

**Perspectives on participation in education of medical students**

| | |
|---|---|
| Would become active in the training of medical students in Siegen | 71 (83%) |
| One-day observation | 68 (80%) |
| Two-week rotation | 63 (74%) |
| clinical elective | 58 (68%) |
| Practical Year | 49 (58%) |
| Lectures at a university | 29 (34%) |
| Participation in research projects | 58 (68%) |
| Recruitment of patients in practice | 48 (57%) |

**Qualification of non-medical staff**

| | |
|---|---|
| Practice nurse | 33 (39%) |
| Number of practice nurses | 1: 65 (77%) |
| | 2: 14 (16%) 4: 6 (7%) |
| Staff member currently doing the practice nurse curriculum | 9 (11%) |
| Number of staff members currently doing the practice nurse curriculum | 1: 48 (56%) |
| | 2: 37 (44%) |
| Staff member planning to do the curriculum | 30 (35%) |

**Notes.**
GP, general practitioner.

**Table 2   Final version of the MoME-Questionnaire (24 items).**

| | | Agree | Slightly agree | Slightly not agree | Not agree |
|---|---|---|---|---|---|
| 1 | I want to contribute in promoting and educating medical students. | 1 | 2 | 3 | 4 |
| 2 | It is my social responsibility to actively participate in the education of medical students. | 1 | 2 | 3 | 4 |
| 3 | I have a mind to educate students and to share my knowledge. | 1 | 2 | 3 | 4 |
| 4 | Educating students is a knowledge exchange where both sides may benefit from. | 1 | 2 | 3 | 4 |
| 5 | Being an "Academic Teaching Practice" (related to a university) enhances the status of my practice. | 1 | 2 | 3 | 4 |
| 6 | Patients feel that I am more qualified if future medical doctors are trained in my practice. | 1 | 2 | 3 | 4 |
| 7 | Being an "Academic Teaching Practice" (related to a university) is publicity for my practice. | 1 | 2 | 3 | 4 |
| 8 | I hope that cooperating with a university facilitates access to evidence based information. | 1 | 2 | 3 | 4 |
| 9 | Cooperating with the university is a good chance to get touch with colleagues and build a network. | 1 | 2 | 3 | 4 |
| 10 | Cooperating with the university increases my chances to find a successor for my own practice. | 1 | 2 | 3 | 4 |
| 11 | Students can spend more time with patients what increases patients' satisfaction. | 1 | 2 | 3 | 4 |
| 12 | Teaching students also means to be up-to-date with respect to medical information. | 1 | 2 | 3 | 4 |
| 13 | Positive experiences I made during my own training period motivate me to participate in the education of medical students. | 1 | 2 | 3 | 4 |
| 14 | Negative experiences I made during my own training period motivate me to participate in the education of medical students. | 1 | 2 | 3 | 4 |
| 15 | I believe I am too old to teach medical students. | 1 | 2 | 3 | 4 |
| 16 | I do not have sufficient didactical competencies. | 1 | 2 | 3 | 4 |
| 17 | Students derange practice administration. | 1 | 2 | 3 | 4 |
| 18 | I can treat less patients if I instruct students in my practice. | 1 | 2 | 3 | 4 |
| 19 | Being exposed to students frequently my patients are less satisfied. | 1 | 2 | 3 | 4 |
| 20 | I operate at full capacity regarding patient treatment. This is why I do not have time to teach and train students in my practice. | 1 | 2 | 3 | 4 |
| 21 | I operate at full capacity regarding patient treatment. This is why I do not have time to teach and train students out of my practice. | 1 | 2 | 3 | 4 |
| 22 | I am not interested in teaching medical students (lectures at the university). | 1 | 2 | 3 | 4 |
| 23 | I am not interested in instructing medical students in my practice. | 1 | 2 | 3 | 4 |
| 24 | Family commitments debar me from participating in teaching students. | 1 | 2 | 3 | 4 |

**Notes.**

Items belonging to factor "commitment".
Items belonging to factor "personal benefit".

**Table 3  Descriptive statistics of the remaining 24 items of the MoME-Q.**

|  | Mean | SD | Median | Skewness | Kurtosis |
|---|---|---|---|---|---|
| Item 1 | 1.46 | .73 | 1 | 1.63 ($p < .001$) | 2.28 ($p < .001$) |
| Item 2 | 1.88 | .75 | 2 | 0.52 ($p = .03$) | −0.08 ($p = .44$) |
| Item 3 | 1.61 | .71 | 1 | 0.93 ($p < .001$) | 0.39 ($p = .23$) |
| Item 4 | 1.46 | .65 | 1 | 1.10 ($p < .001$) | 0.11 ($p = .42$) |
| Item 5 | 2.12 | .84 | 2 | 0.36 ($p = .09$) | −0.42 ($p = .22$) |
| Item 6 | 2.42 | .86 | 2 | 0.07 ($p = .39$) | −0.61 ($p = .13$) |
| Item 7 | 2.44 | .91 | 2 | 0.00 ($p = .50$) | −0.76 ($p = .08$) |
| Item 9 | 2.55 | .82 | 3 | 0.05 ($p = .43$) | −0.51 ($p = .17$) |
| Item 10 | 2.06 | .86 | 2 | 0.45 ($p = .04$) | −0.45 ($p = .20$) |
| Item 11 | 2.01 | .88 | 2 | 0.66 ($p = .007$) | −0.12 ($p = .42$) |
| Item 13 | 2.39 | .77 | 2 | 0.31 ($p = .12$) | −0.18 ($p = .37$) |
| Item 14 | 1.85 | .75 | 2 | 0.77 ($p = .002$) | 0.80 ($p = .07$) |
| Item 16 | 2.13 | 1.02 | 2 | 0.35 ($p = .09$) | −1.08 ($p = .02$) |
| Item 17 | 2.62 | 1.01 | 3 | −0.17 ($p = .26$) | −1.04 ($p = .03$) |
| Item 18 | 3.58 | .73 | 4 | −1.97 ($p < .001$) | 3.87 ($p < .001$) |
| Item 19 | 3.19 | .84 | 3 | −0.74 ($p = .003$) | −0.20 ($p = .36$) |
| Item 20 | 2.85 | .85 | 3 | −0.29 ($p = .14$) | −0.56 ($p = .15$) |
| Item 21 | 2.32 | .69 | 2 | 0.14 ($p = .30$) | −0.08 ($p = .44$) |
| Item 22 | 2.96 | .68 | 3 | −0.42 ($p = .06$) | 0.54 ($p = .16$) |
| Item 23 | 2.85 | .87 | 3 | −0.37 ($p = .08$) | −0.47 ($p = .19$) |
| Item 24 | 2.34 | 1.13 | 2 | 0.25 ($p = .17$) | −1.32 ($p = .007$) |
| Item 26 | 3.44 | .88 | 4 | −1.31 ($p < .001$) | 0.46 ($p = .19$) |
| Item 27 | 3.58 | .75 | 4 | −1.77 ($p < .001$) | 2.47 ($p < .001$) |
| Item 28 | 3.09 | 1.03 | 3 | −0.79 ($p = .001$) | −0.62 ($p = .12$) |

The remaining items were hypothesized to load on the 4 different factors conviction, personal benefit, personal resources, and time management. The confirmatory factor analysis with the robust ULSMV estimation method showed a good model fit: $\chi^2/df = 1.36$, $RMSEA = .066$, $SRMR = .096$. Factors conviction and personal resources correlated .97 and there was also a high correlation between factors conviction and time management ($r = -.86$). Model parsimony is a main target in confirmatory factor analysis. Highly correlating factors do not convey additional information. Therefore, factors conviction, personal resources, and time management were united into one factor called "commitment" and a new analysis postulating a two-factor model ("commitment" and "personal benefits") was performed. The confirmatory factor analysis with the robust ULSMV estimation method again showed a good model fit: $\chi^2/df = 1.38$, RMSEA = .067, SRMR = .098.

As shown in Table 4, all factor loadings are in the satisfactory range and the correlation between the two factors is also acceptable ($r = .503$). Therefore, this two-factor solution can be accepted and we calculated two subtest scores in the MoME-Q for further analyses. We have to mention that this is not a confirmatory but a model generating approach (*Jöreskog, 1993*) which means that this structure has to be confirmed in a new sample. This was done as the instrument was newly developed and there were only preliminary hypotheses about

**Table 4  Factor loadings of the two-factor solution in the confirmatory factor analysis.**

|  | Factor loading |
|---|---|
| Factor commitment | |
| Item 1 | .78 |
| Item 2 | .64 |
| Item 3 | .85 |
| Item 4 | .67 |
| Item 16 | .54 |
| Item 17 | .39 |
| Item 18 | −.54 |
| Item 19 | −.56 |
| Item 20 | −.57 |
| Item 21 | −.36 |
| Item 22 | −.48 |
| Item 23 | −.84 |
| Item 24 | −.53 |
| Item 26 | −.64 |
| Item 27 | −.81 |
| Item 28 | −.53 |
| Factor personal benefit | |
| Item 5 | .58 |
| Item 6 | .66 |
| Item 7 | .53 |
| Item 9 | .51 |
| Item 10 | .80 |
| Item 11 | .46 |
| Item 13 | .43 |
| Item 14 | .70 |

the structure of the questionnaire. After reversing items 18, 19, 20, 21, 22, 23, 24, 26, 27, and 28 the scale score of "commitment" was calculated.

The mean of the scale "commitment" was 31.0 (SD 8.4) with a median of 30, a minimum of 19, and a maximum of 53 (Fig. 1). Its distribution deviated significantly from a normal distribution: Shapiro–Wilk-Test, $p < .001$; Skewness, $p = .003$; Kurtosis, $p = .35$. Cronbach-$\alpha$ coefficient was .90, omega coefficient was .91, the average inter-item-correlation was .37. Discriminatory power of the items ranged from .31–.79. All values can be classified as satisfactory to high.

The mean of the scale personal benefit was 17.9 (SD 4.4) with a median of 18, a minimum of 10, and a maximum of 31 (Fig. 2). Its distribution mainly deviated significantly from a normal distribution: Shapiro–Wilk-Test, $p = .02$; Skewness, $p = .02$; Kurtosis, $p = .19$. Cronbach-$\alpha$ coefficient was .81, omega coefficient was also .81, the average inter-item-correlation was .34. Discriminatory power of the items ranged from .33–.65. All values can be classified as satisfactory to reasonably high.

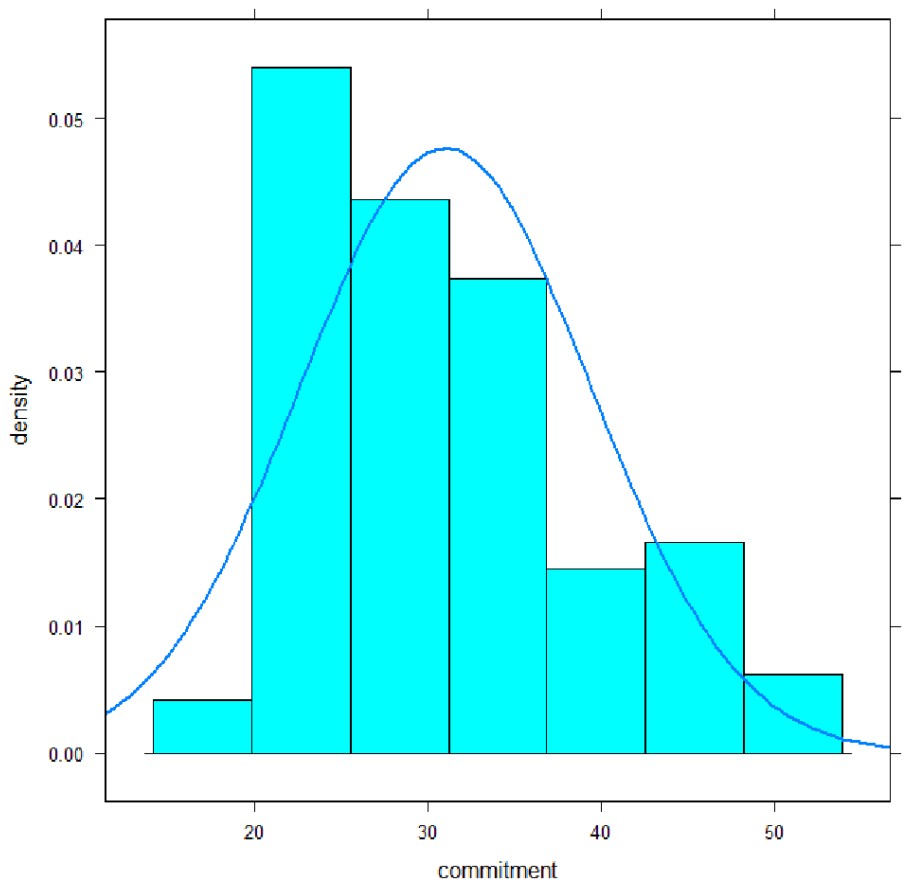

**Figure 1** Distribution of the commitment subscale scores of MoME-Q.

## Association with demographic characteristics

We median dichotomized age of physicians and then compared the two groups on the scales of the MoME-Q. The descriptive values are displayed in Table 5.

There was no significant multivariate effect ($T^2(2, 81) = 4.15, p = .14$). The two age groups did not differ significantly on the scales of the MoME-Q.

We compared those physicians who had an authorization for performing practical education for future GPs with those who did not have this authorization on the scales of the MoME-Q. The descriptive values are displayed in Table 6.

There was a significant multivariate effect ($T^2(2, 82) = 7.32, p = .03$). Univariate analyses revealed a significant difference between the two groups on the subscale personal benefit $t(75.6) = -2.62, p = .01$. A medium effect was shown by effect size Cohen's d with .57. Those with authorization to perform practical education for future GPs hope to have more personal benefits than those who do not possess this authorization. No significant difference occurred on the scale commitment $t(78.2) = -0.60, p = .55, d = .13$.

We further compared those practices with a specialized practice nurse or with a practice nurse still in training with those who do not have a specialized practice nurse and who do not intend to have one in the future on the scales of the MoME-Q. This was done

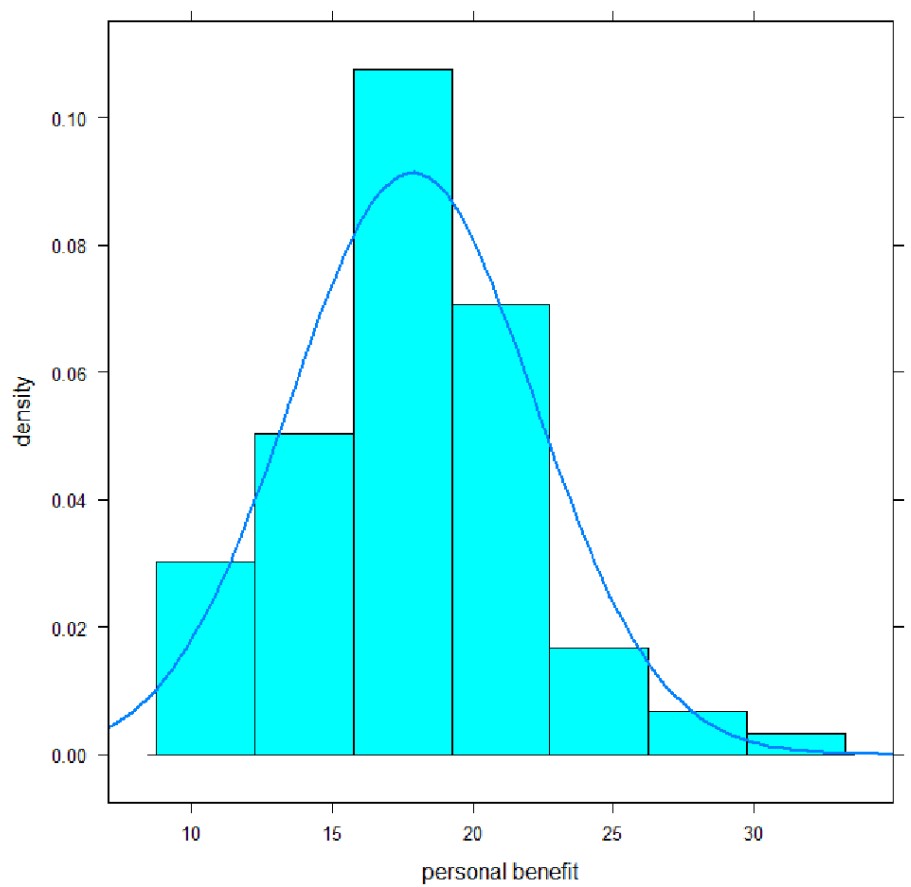

**Figure 2** Distribution of the personal benefit subscale scores of MoME-Q.

**Table 5** Descriptive values of the MoME-Q scales split by age groups.

|  | Commitment mean (sd) | Personal benefit mean (sd) |
|---|---|---|
| age <median ($n = 42$) | 29.8 (7.4) | 16.9 (4.3) |
| age ≥ median ($n = 42$) | 31.8 (9.1) | 18.8 (4.4) |

**Table 6** Descriptive values of the MoME-Q scales split by authorization for performing practical education for GPs.

|  | Commitment mean (sd) | Personal benefit mean (sd) |
|---|---|---|
| Authorization ($n = 45$) | 30.5 (7.9) | 16.7[**] (3.8) |
| No authorization ($n = 40$) | 31.6 (9.0) | 19.1[**] (4.7) |

**Notes.**
[**]$p = .01$.

**Table 7** Descriptive values of the MoME-Q scales split by practices with and without a specialized practice nurse.

|  | Commitment mean (sd) | Personal benefit mean (sd) |
|---|---|---|
| No practice nurse ($n = 45$) | 33.4* (8.5) | 18.3 (4.1) |
| Practice nurse ($n = 39$) | 28.4* (7.6) | 17.4 (4.8) |

Notes.
*$p = .02$.

**Table 8** Descriptive values of the MoME-Q scales split by physicians with and without experience in medical education.

|  | Commitment mean (sd) | Personal benefit mean (sd) |
|---|---|---|
| experience ($n = 50$) | 27.6*** (6.4) | 16.8 (3.6) |
| no experience ($n = 35$) | 35.8*** (8.7) | 19.3 (5.0) |

Notes.
***$p < .0001$.

to investigate if there is a trend to support education of staff members in general. The descriptive values are displayed in Table 7.

There was a significant multivariate effect ($T^2(2, 81) = 7.93, p = .02$). Univariate analyses revealed a significant difference between the two groups on the subscale commitment: $t(81.9) = 2.82, p = .006$. A medium effect was shown by effect size $d = .62$. Those physicians with a practice nurse show a higher commitment than those physicians who do not have a practice nurse and who do not have the intention to have a practice nurse in the future. There was no significant difference between the two groups on the scale personal benefit: $t(75.8) = 0.85, p = .40$ with a small effect size of $d = .19$.

We compared those physicians with experience in medical education with those who did not have experience in medical education on the scales of the MoME-Q. The descriptive values are displayed in Table 8.

There was a significant multivariate effect ($T^2(2, 82) = 25.93, p < .0001$. Univariate analyses revealed a significant difference between the two groups on the subscale commitment: $t(59.2) = -4.74, p < .0001$. A large effect was shown by effect size $d = 1.05$. Those physicians with experience in medical education expressed a significantly higher commitment for medical education than those physicians without experience in medical education. There was also a significant difference between the two groups on the scale personal benefit: $t(57.8) = -2.50, p = .015$ with a medium effect size of $d = .55$. Those physicians with experience in medical education expect a higher personal benefit from medical education than those physicians who do not have experience in medical education.

## Associations with burnout and work satisfaction

We calculated Spearman correlation coefficients between MoME-Q scales and subscales of the Maslach Burnout Inventory (MBI) (Table 9).

The MoME-Q subscale "commitment" correlated significantly with all three subscales of the MBI. Correlations were in the medium range around |.30|. Those physicians with higher commitment scores and therefore lower commitment to teach had higher scores on emotional exhaustion and depersonalization and vice versa. Those physicians with lower

**Table 9  Spearman correlations of the MoME-Q scales with subscales of the Maslach Burnout Inventory (MBI).** Due to multiple testing the significance level had to be adjusted to $p = .05/6 = .008$.

| MBI MoME-Q | Emotional exhaustion | Depersonalization | Personal accomplishment |
|---|---|---|---|
| Commitment | .30 | .33 | −.35 |
| | $p = .006$ | $p = .001$ | $p = .002$ |
| Personal benefit | .08 | .03 | −.08 |
| | $p = .44$ | $p = .75$ | $p = .46$ |

**Table 10  Spearman correlations of the MoME-Q scales with subscales of the Work Satisfaction Questionnaire.** Due to multiple testing the significance level had to be adjusted to $p = .05/12 = .004$.

| MoME-Q | Patient care | Burden | Income-prestige | Personal rewards | Professional relations | Global item |
|---|---|---|---|---|---|---|
| Commitment | −.19 | .01 | .01 | −.15 | −.10 | −.25 |
| | $p = .08$ | $p = .95$ | $p = .93$ | $p = .17$ | $p = .37$ | $p = .02$ |
| Personal benefit | −.03 | .09 | .07 | −.09 | .02 | −.03 |
| | $p = .76$ | $p = .44$ | $p = .53$ | $p = .42$ | $p = .83$ | $p = .76$ |

commitment scores and therefore higher commitment to teach had higher scores on the MBI subscale personal accomplishment and vice versa. The correlations of the MoME-Q subscale "personal benefits" with MBI subscales were around zero.

None of the correlations between the MoME-Q subscales and the subscales of the Work Satisfaction Questionnaire reached significance after Bonferroni correction (Table 10). The only tendency which can be reported is that those physicians with lower commitment scores and therefore higher commitment to teach are more satisfied with their current job situation and vice versa. A German version of the MoME-Q can be found in the supplement (Table S2).

# DISCUSSION

We present a 24-item questionnaire to assess motivation in non-experienced GPs to participate in the education of medical students. After taking a model generating approach in confirmatory factor analysis, the MoME-Q could be best characterized by a two-factor model instead of the initial hypothesis of a four-factor structure. Factor "commitment" consisted of 16 items with Cronbach- $\alpha$ and omega-coefficients around .90 while factor "personal benefit" had eight items with Cronbach- $\alpha$ and omega-coefficients being around .80.

Mean differences between most groups based on demographic characteristics demonstrate the validity of the MoME-Q to highlight relevant aspects for the motivation in medical education. Median dichotomized age groups did not differ significantly in their scores on the MoME-Q subscales. Physicians with authorization to train GP residents expect to have more personal benefits than those who do not possess this authorization (medium effect size). Physicians with a practice nurse show a higher commitment to teach than those physicians who do not have a practice nurse and who do not have the intention to have a practice nurse in the future (medium effect size). The latter two findings

show that GPs who set value on further education of their employees also tend to teach medical students. Physicians with experience in medical education expressed a significantly higher commitment for medical education and expect higher personal benefit than those physicians without experience in medical education (large and medium effect sizes, respectively). This finding is quite promising as it shows that GP teachers see the benefit from their efforts and are apt to continue medical education of students. Correlations with the Maslach Burnout Inventory revealed that physicians with higher commitment scores and therefore lower commitment to teach had higher scores on emotional exhaustion and depersonalization and vice versa, which is not surprising. Physicians with lower commitment scores and therefore higher commitment to teach had higher scores on the MBI subscale personal accomplishment and vice versa. Work satisfaction surprisingly was not significantly associated with motivation to teach. The only tendency which can be reported is that those physicians with lower commitment scores and therefore higher commitment to teach are more satisfied with their current job situation and vice versa.

The results of our study are promising regarding the project to establish a medical campus in Siegen. Much more GPs than expected are willing to participate in the medical education process of students, especially compared to the current situation. A total of 83% of the GPs can imagine participating in medical education. Hence, the willingness to have students in the practice for short terms (i.e., two-week rotations) is higher than for long-term education (i.e., practical year). Teaching students out of the own practice is not wanted by the majority of GPs as this would mean an additional expenditure of time. Nevertheless, these results need to be interpreted with caution. Being motivated according to the questionnaire needs to be turned into action, i.e., participation in training programs and educating students in reality. Furthermore, the results also show that the willing population of GPs needs preparation and support, e.g., by didactical courses and training programs. In addition, delegation plays a growing role in patient care, especially in rural areas. Courses for non-medical staff (doctor's assistants) are available to become practice nurses that take on more responsibility in patient care. This is becoming more relevant and important as due to a shortage of GPs delegation gets more important. Although investigating a rural area, less than half of the GPs do have practice nurses and what even surprises more is that only 11% of the GPs do have staff members in training to become practice nurses. The reasons for that remain unclear. We can only speculate that the absence of a course offer might lead to this low rate of staff members in training. Currently, courses are only available in a distance of about 100 km.

Looking at research, the majority can imagine to participate in specific research projects. They are basically also willing to recruit patients in the practice, which must be seen as the basis for primary care research.

A strength of our study was that we were able to conduct a full population survey in a limited geographic area. The response rate of 53.8% can be regarded as satisfactory as the survey contained several self-relevant questions regarding own future work prospects and continuity of practices. These are topics which might cause psychological irritation and might therefore be avoided. This might in turn result in a rejection to participate in a survey containing questions having a possible negative influence on self-esteem (*Harmon-Jones*

& Harmon-Jones, 2007). Of course, our study is subject to some limitations. First, the sample size for confirmatory factor analysis was smaller than the proposed $n = 200$ in the literature (*Brown, 2015*) although the resulting solutions had satisfactory quality criteria. Our results regarding the development of the MoME-Q should be replicated with independent and larger samples as we applied a model generating approach in confirmatory factor analysis. This means that the proposed two factorial solution has to be confirmed with different samples in order to be called a stable solution (*Jöreskog, 1993*). Second, GPs were investigated and our conclusions should therefore be restricted to this specific group.

As already mentioned above, there is no appropriate tool available to assess the motivation of General Practitioners to teach medical students for the purpose of our study. The study by Thomson et al., where we derived some items from, is a qualitative study. The study group undertook semi-structured interviews with GPs, who do already have teaching experiences (*Thomson et al., 2014*). In the study of *Ingham et al. (2015)*, Australian GPs who are already functioning as GP supervisors were investigated by means of semi-structured interviews. The article of Dybowski et al. presents the validation of the Physician Teaching Motivation Questionnaire (PTMQ). This questionnaire is more appropriate to assess motivation of physicians who are already involved in teaching, which makes an important difference as we focus on physicians with no or almost no teaching experience. The validation was further done at a study group of hospital-based-physicians who work at university hospitals (*Dybowski & Harendza, 2015*), whereas we look at GPs who work in their private practices and who mostly had no prior experiences in teaching medical students.

## CONCLUSION

We for the first time present an instrument to assess motivation of GPs with less or no teaching experience to take part in the medical education of students. We could demonstrate that the MoME-Q is an appropriate tool to measure motivation for teaching participation of GPs. Motivation is a complex construct, which is subject to many different influencing factors such as work satisfaction and prior experiences. Future studies with larger number of GPs should be carried out to validate and confirm our findings. Whether the MoME-Q is also appropriate for other specialties should also be shown in further empirical studies.

The results of our study are also promising regarding the project to establish a medical campus in a rural region. The problem to get a sufficient number of GPs involved in teaching purposes to face the challenge of the masterplan 2020 seems solvable. Nevertheless, the results also show that the willing population of GPs needs preparation and support, e.g., by didactical courses and training programs.

**List of abbreviations**

| | |
|---|---|
| **CFI** | Comparative Fit Index |
| **GP** | General Practitioner |
| **HaMEdSi** | Hausärzte (GPs) for Medical EDucation in Siegen-Wittgenstein |
| **kNN** | k nearest neighbor algorithm |
| **MBI** | Maslach Burnout Inventory |

| MoME-Q | Motivation of Medical Education questionnaire |
|---|---|
| **PTMQ** | Physician Teaching Motivation Questionnaire |
| **SD** | standard deviation |
| **SRMR** | standardized root mean square residual |
| **TLI** | Tucker Lewis Index |
| **ULSMV** | Unweighted Least Squares Estimator |

## ACKNOWLEDGEMENTS

We thank all GPs who participated in this study without receiving financial compensation.

### Funding

The authors received no funding for this work.

### Competing Interests

The authors declare there are no competing interests.

### Author Contributions

- Charles Christian Adarkwah and Annette Schwaffertz conceived and designed the experiments, performed the experiments, analyzed the data, contributed reagents/materials/analysis tools, prepared figures and/or tables, authored or reviewed drafts of the paper, approved the final draft.
- Joachim Labenz conceived and designed the experiments, performed the experiments, contributed reagents/materials/analysis tools, prepared figures and/or tables, authored or reviewed drafts of the paper, approved the final draft.
- Annette Becker conceived and designed the experiments, contributed reagents/materials/analysis tools, prepared figures and/or tables, authored or reviewed drafts of the paper, approved the final draft.
- Oliver Hirsch conceived and designed the experiments, analyzed the data, contributed reagents/materials/analysis tools, prepared figures and/or tables, authored or reviewed drafts of the paper, approved the final draft.

### Human Ethics

The following information was supplied relating to ethical approvals (i.e., approving body and any reference numbers):

The study was performed in accordance with the Declaration of Helsinki and approved by the research ethics committee of the University of Marburg (Az.: Studie 127/17).

### Data Availability

Adarkwah, Charles Christian; Schwaffertz, Annette; Labenz, Joachim; Becker, Annette; Hirsch, Oliver (2018): HaMEdSi study. figshare. Dataset. https://doi.org/10.6084/m9.figshare.7357034.v1.

## Supplemental Information

Supplemental information for this article can be found online at http://dx.doi.org/10.7717/peerj.6235#supplemental-information.

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
