# Peer review of "GPs’ motivation for teaching medical students in a rural area—development of the Motivation for Medical Education Questionnaire (MoME-Q)"

_PeerJ, doi:10.7717/peerj.6235_

## Round 0.1 · original submission · Major Revisions

Thank you very much for your submission. Reviewers indicated substantial revisions are needed before a decision on publication is made. Please consider and respond to all reviewer suggestions. Thank you.

Reviewer 1 ·

Basic reporting

Generally well written but there are a number of grammatical errors (eg ln 82 has/have), spelling errors (eg ln 82 retierement), poor word choice (ln 58 “shortness” rather than “shortage”) and problems with punctuation (eg ln 254 quote marks). Introduction and background are relevant. Table layout could be improved eg for Table 1: include “n” (with % in brackets); is it necessary to include both %Yes and %No when total are always 100%? Suggest that Tables 4, 5, 6 and 7 are combined into a single table.

Experimental design

Experimental design is adequate and well described. Include “n” for number of GPs invited to take part (ln 118), and number that attended informative meeting (ln 131). Normal terminology is “slightly agree” and “slightly not agree” instead of “rather agree” and “rather not agree”.

Validity of the findings

I do not have the expertise to comment on the appropriateness of the statistical tests chosen. The questionnaire that has been developed addresses a clearly identified need, and appears to yield valid results.

Additional comments

No additional comments

Reviewer 2 ·

Basic reporting

The manuscript is written in a clear language with few minor weaknesses (see below) but should be improved in details. In introduction, statements should be proved with (current literature) references consequently.
* * *
- Please add absolute numbers where % is stated.

- Line 61: sentence "Furthermore….) is hard to understand. Please rewrite with less commas.
-Line 55/59: reference missing
- Line 64.66: please correct: three months in ambulant care (not only General practice).
- please check spelling in references (line 581, capital letters)

- please check tables and add legends
- in table 1 age is missing. Please add absolute numbers and missing values
- in table 1: practice size can not be correlated with single/group. Please change into "practice form"
- table 1: please add information about the difference practice assitant/practice nurse
- please add the questionnaire as a table, or define the items in table 2. These Information is crucial and should not be reserved to supplements.
- table 2: is this right with 28 items? why not 24?

Experimental design

The research question (line 111-114) should be focussed and adapted to the abstract (construct and test a questionnaire) and the title. Which is the main question - "to develop a questionnaire" (abstract, title) or "to figure out motivation) (line 111) -??

The rationale should then focus more on the research question. Information on known motivational factors for teaching for physiscians and General Practitioners should be added, with literature references. In Line 99-104 I miss the context, and expect factors on teaching motivation and why a questionnaire for GPs is needed in addition to the existing instruments (add further information after line 105, maybe from the discussion line 442 ff)

Some other information shows less context to the research question (line 84-86, 92-96) and is expendable.

The Methods section needs more details, and more focus on the research question. Please check the whole section for any potential relevant information, i.e.:
- line 117: From which source did you get the contacts? Were all GPs invited, or all practices?
- line 120: what is meant by work perspective?
- line 131: how many took part?
- line 149-153: please explain further, please cite which literature is meant
- missing data: how did you handle missing data?

Further Instruments: What is the context of this questionnaires ? Are they important for the development of the questionnaire?

Development of questionnaires:
-Please provide information on the items: which items derived from the literature and from the discussion rounds?
- Please provide more information why you added qualification of non medical staff.

Results:
- Line 211: Please start with numbers of GPs invited and number of participants
- please show data on "missing data"

Discussion:
-please re-write the first paragraph stating the relevant results corresponding to the research question.
-Please add findings from the literature on your statements line 397-402. Are these results known before?
- In total, the discussion should focus more on relevant findings regarding the research questions (based on existing literature) .
- line 404 following: please be more careful with the interpretation, in particular since data are based on motivation. I miss the reflection of the a gap between motivation, participation in training programs and educating students in reality.

- line 415 ff: the context to the research question is not fully clear. If staff education might play a role, please explain before why you take this into consideration, and how staff education works in Germany. In the discussion section please stay focused - is line 417 -422 really important, and why?

limitations:
- development of the questionnaire is based on a GP population in a rural area with shortness of physicians. For further research, the questionnaire should be validated with a population of urban GPs. Are there any known results from other authors?

Validity of the findings

Novelty is assessed in line 456 - since PeerJ explicitely wants to avoid statements on impact and novelty, this seems expendable.

Please limit the conclusions to research questions and results, and avoid speculative statements (line 466, be careful of the motivation - reality-gap)

Additional comments

With developing a motivation questionnaire for General Practitioners, you treat a relevant theme on a solide methodological base, and present results in a clearly written manuscript.
However, quite a number of more or less major/minor neglects (see above) impair the scientific presentation substantially.
Please revise the whole manuscript intensively. Beyond the quoted examples, rationales and details (references, numbers) should be supplemented, and the whole text and all tables should be checked carefully regarding completeness and comprehensibility.

Reviewer 3 ·

Basic reporting

Good literature review and description of findings apart from abstract. Results and conclusion section needs re-writing. E.g., authors mention "Correlations with the MBI were in the expected directions" without providing any detail in results section and then mention factors in the conclusion which were not mentioned in results.
It is suggested to highlight the salient takeaway from the study in the abstract and conclusion be based on the described results in the abstract.

Experimental design

Research question is well defined, relevant and meaningful and within the scope of journal. Methods have been described in detail. However, readers would want to know more about the factors affecting motivation so as to see if anything can be done to enhance this. Detailed analysis with discussion of the factors need to be added so that research can help in improving the motivation levels in the long term.

Validity of the findings

Conclusions of the article are well stated and supported by data, however, same is not true of the abstract conclusion which would need to be improved as stated above.

---

## Round 0.2 · accepted · Accept

Please note the remarks of one of the reviewers and make suggested revisions to the abstract while in production. Thank you.

Reviewer 2 ·

Basic reporting

The manuscript improved, still, please add absolute numbers in the abstract (results).

Experimental design

The manuscript improved, in particular, the research question is more focused now and relvant information has been added.

Validity of the findings

The manuscript improved.

Reviewer 3 ·

Basic reporting

Authors have made the necessary changes and manuscript is suitable for publication now.

Experimental design

Authors have made the necessary changes and manuscript is suitable for publication now.

Validity of the findings

Authors have made the necessary changes and manuscript is suitable for publication now.

---

## Author Rebuttal · Round 0.2

Lebenswissenschaftliche Fakultät
**Lehrstuhl für Versorgungsforschung**
Dr. Dr. med. Charles Christian Adarkwah, M.Sc.
- Vertretungsprofessor -

UNIVERSITÄT SIEGEN ● Fakultät V ● 57068 Siegen
Editorial Office

Weidenauer Str. 167
57076 Siegen
Telefon  +49 271 740-5131
charles.adarkwah@uni-siegen.de
www.uni-siegen.de

Siegen,  hier Datum (Bsp.: 1. Oktober 2006)

**Our manuscript "GPs' motivation for teaching medical students in a rural area - development of the Motivation for Medical Education Questionnaire (MoME-Q)" – submission of revised version**

Dear Editors,

We would like to thank for valuable reviewer comments on our manuscripts.
We discussed all issues in detail and incorporated the reviewers' suggestions into the new version of our manuscript. Changes are indicated with track changes.

We feel that the manuscript is now suitable for publication in PeerJ and are looking forward to your response.

Best regards on behalf of the authors,

Charles Christian Adarkwah, MD, PhD

# Reviewer 1

**Basic reporting**
Generally well written but there are a number of grammatical errors (eg ln 82 has/have), spelling errors (eg ln 82 retierement), poor word choice (ln 58 "shortness" rather than "shortage") and problems with punctuation (eg ln 254 quote marks). Introduction and background are relevant. Table layout could be improved eg for Table 1: include "n" (with % in brackets); is it necessary to include both %Yes and %No when total are always 100%? Suggest that Tables 4, 5, 6 and 7 are combined into a single table.
**Response: We thank the reviewer for the positive overall judgement.**
**The manuscript was corrected prior resubmission. Layout of Table 1 was adjusted: the %No were deleted. We now present the total "n" in the table as well. For a better understanding we would like to keep tabled 4-7 separate items so that the respective table can be displayed in close relation to the text passage.**

**Experimental design**
Experimental design is adequate and well described. Include "n" for number of GPs invited to take part (ln 118), and number that attended informative meeting (ln 131). Normal terminology is "slightly agree" and "slightly not agree" instead of "rather agree" and "rather not agree".
**Response: We followed the reviewer's recommendation. The adjustments suggested were made to the relevant section of the manuscript.**

**Validity of the findings**
I do not have the expertise to comment on the appropriateness of the statistical tests chosen. The questionnaire that has been developed addresses a clearly identified need, and appears to yield valid results.
**Response: Thank you very much!**

**Comments for the author**
No additional comments

## Reviewer 2

**Basic reporting**
The manuscript is written in a clear language with few minor weaknesses (see below) but should be improved in details. In introduction, statements should be proved with (current literature) references consequently.
**Response: We first of all thank the reviewer for an extensive review of our manuscript. Literature was added to the introduction to prove the statements made.**
* * *
- Please add absolute numbers where % is stated.
**Response: Absolute numbers were added to table 1 for a better understanding (see statemen of reviewer 1).**

- Line 61: sentence "Furthermore….) is hard to understand. Please rewrite with less commas.
**Response: The sentence was corrected.**

-Line 55/59: reference missing
**Response: A reference was added to the manuscript.**

- Line 64.66: please correct: three months in ambulant care (not only General practice).
**Response: We thank the reviewer for this remark and corrected the sentence. One could speculate that the majority of students will choose General Practice as this will become mandatory in the examination.**

- please check spelling in references (line 581, capital letters)
**Response: Adjusted.**

- please check tables and add legends
**Response: We added legends as far as there was a need to explain abbreviations.**

- in table 1 age is missing. Please add absolute numbers and missing values
**Response: The values were added to table 1.**

- in table 1: practice size can not be correlated with single/group. Please change into "practice form"
**Response: The table was adjusted according to the recommendation.**

- table 1: please add information about the difference practice assitant/practice nurse
**Response: Table 1 only mentions practice nurses. We do not talk about "practice assistants" in this context at all. In the discussion section we state that "Courses for non-medical staff (doctor's assistants) are available to become practice nurses that take on more responsibility in patient care.", which implies that there is an option to educate oneself to become practice nurse.**

- please add the questionnaire as a table, or define the items in table 2. These Information is crucial and should not be reserved to supplements.
**Response: We moved supplementary table 2 (24 items) to the main part of the manuscript. The initial (28 items) and the modified (24 items) questionnaire are still available as supplementary material.**

- table 2: is this right with 28 items? why not 24?
**Response: In this table, only 24 items are mentioned. The initial item numbers are attached. To avoid confusion we now present the final item numbers (in line with table 2) in**

the first column, and the initial item numbers in the second column (in line w/ supplementary tables 1 and 2).

**Experimental design**
The research question (line 111-114) should be focussed and adapted to the abstract (construct and test a questionnaire) and the title. Which is the main question - "to develop a questionnaire" (abstract, title) or "to figure out motivation) (line 111) -??
**Response:We thank the reviewer for this comment. Our research question indeed has to parts. This is because to figure out GP's motivation we had to develop a questionnaire as no appropriate tool is available. In this case, one cannot go without the other. The abstract was adjusted accordingly.**

The rationale should then focus more on the research question. Information on known motivational factors for teaching for physicians and General Practitioners should be added, with literature references. In Line 99-104 I miss the context, and expect factors on teaching motivation and why a questionnaire for GPs is needed in addition to the existing instruments (add further information after line 105, maybe from the discussion line 442 ff)
**Response: The context was now more clarified and a statement added: "Having motivated teachers does have an impact on students' performance."**
**We now stress that for our population no instrument is available to assess motivation. Initial line 442ff more presents a discussion of the literature which we would like to keep in the discussion section of our manuscript.**

Some other information shows less context to the research question (line 84-86, 92-96) and is expendable.
**Response: The reviewer is right in this point that the passage shows less context to the research question. But we consider this information important at least for those readers, who are not in detail familiar with the GP shortage in various European countries. We would therefore like to keep this straightforward information in the manuscript.**

The Methods section needs more details, and more focus on the research question. Please check the whole section for any potential relevant information, i.e.:
- line 117: From which source did you get the contacts? Were all GPs invited, or all practices?
**Response: Information about the source was added; as mentioned all GPs were invited.**

- line 120: what is meant by work perspective?
**Response: The sentence was rephrased in order to clarify the meaning:**
**"In addition, they were asked in detail regarding their future prospects of working as a GP."**

- line 131: how many took part?
**Response: 45 GPs took part. This was now added to the manuscript.**

- line 149-153: please explain further, please cite which literature is meant
**Response: We now mention the literature we refer to again in this context. Please see initial lines 137-139)**

- missing data: how did you handle missing data?
**Response: There were up to 3 missing values on single items of the MoME-Q. They were replaced by the k nearest neighbor algorithm (kNN). There were 1-2 missing values on variables which were not decisive for the central analyses. Therefore, these could be ignored also in accordance with the literature on missing value analysis.**

**This was now indicated in the manuscript.**

Further Instruments: What is the context of this questionnaires ? Are they important for the development of the questionnaire?
**Response: These questionnaires were used to external validation. We assessed and present associations between the motivation to take part in medical education and burnout risk as well as work satisfaction.Thus, it is crucial to at least give a short overview on the respective instrument.**

Development of questionnaires:
-Please provide information on the items: which items derived from the literature and from the discussion rounds?
**Response: We thank the reviewer for this comment. It is difficult, more precisely impossible to make a clear distinction. This is because the literature we refer to mainly represents qualitative research results. No clear item definition was made in those articles so that ideas from the articles and from discussion rounds merged together and resulted in clear item definitions. Therefore it is not possible to clearly assign items.**

- Please provide more information why you added qualification of non medical staff.
**Response: This was done to investigate if there is a trend to support education of staff members. All GPs do have non-medical staff, whereas most GPs do not have medical staff members. This was now added to the result section of the manuscript.**

Results:
- Line 211: Please start with numbers of GPs invited and number of participants
**Response: Exactly this is what we now mention here. In the method section we explain that all GPs were invited to take part in the study.**

- please show data on "missing data"
**Response: There were up to 3 missing values on single items of the MoME-Q. They were replaced by the k nearest neighbor algorithm (kNN). There were 1-2 missing values on variables which were not decisive for the central analyses. Therefore, these could be ignored also in accordance with the literature on missing value analysis. This was now indicated in the manuscript.**

Discussion:
-please re-write the first paragraph stating the relevant results corresponding to the research question.
**Response: This passage was modified. We now start with the first research question concerning the questionnaire development. Results are briefly summarized and discussed. This is followed by the second research question regarding the assessment of motivation (para3).**

-Please add findings from the literature on your statements line 397-402. Are these results known before?
**Response: We here present a summary of our findings. To the best of our knowledge, this is not represented in the literature as this kind of association has not been investigated before.**

- In total, the discussion should focus more on relevant findings regarding the research questions (based on existing literature)

**Response: Please allow as to refer to our statement regarding the experimental design (see above). We discussed the relevant findings both ways. Literature on this topic is scarce and was discussed to the best of our knowledge.**

- line 404 following: please be more careful with the interpretation, in particular since data are based on motivation. I miss the reflection of the a gap between motivation, participation in training programs and educating students in reality.
**Response: We thank the reviewer for this valuable comment. This was now added to the discussion section of the manuscript: "Nevertheless, these results need to be interpreted with caution. Being motivated according to the questionnaire needs to be turned into action, i.e. participation in training programs and educating students in reality."**

- line 415 ff: the context to the research question is not fully clear. If staff education might play a role, please explain before why you take this into consideration, and how staff education works in Germany. In the discussion section please stay focused - is line 417 -422 really important, and why?
**Response: We now also mention, why staff education is important, especially in rural regions. "This is becoming more relevant and important as due to a shortage of GPs delegation gets more important."**

limitations:
- development of the questionnaire is based on a GP population in a rural area with shortness of physicians. For further research, the questionnaire should be validated with a population of urban GPs. Are there any known results from other authors?
**Response: The answer to this question is no. Our instrument can also be used for urban GPs as we do not have incorporated items that limit the validity to GPs in rural area. We plan to further validate the questionnaire using other populations, also including more urban areas.**

**Validity of the findings**
Novelty is assessed in line 456 - since PeerJ explicitly wants to avoid statements on impact and novelty, this seems expendable.
**Response: In fact, that exactly is, what our study was meant to do. We think that this statement is crucial also to justify our scope of research.**

Please limit the conclusions to research questions and results, and avoid speculative statements (line 466, be careful of the motivation - reality-gap)
**Response: The sentence was deleted.**

**Comments for the author**
With developing a motivation questionnaire for General Practitioners, you treat a relevant theme on a solide methodological base, and present results in a clearly written manuscript.
However, quite a number of more or less major/minor neglects (see above) impair the scientific presentation substantially.
Please revise the whole manuscript intensively. Beyond the quoted examples, rationales and details (references, numbers) should be supplemented, and the whole text and all tables should be checked carefully regarding completeness and comprehensibility.
**Response: We thank the reviewer for valuable feedback on our manuscript and hope that it now meets all criteria to be published in PeerJ.**

# Reviewer 3

**Basic reporting**

Good literature review and description of findings apart from abstract. Results and conclusion section needs re-writing. E.g., authors mention "Correlations with the MBI were in the expected directions" without providing any detail in results section and then mention factors in the conclusion which were not mentioned in results.

It is suggested to highlight the salient takeaway from the study in the abstract and conclusion be based on the described results in the abstract.

**Response: We thank the reviewer for this positive overall statement. The abstract was revised according to the reviewer's suggestions. The correlations with the MBI are provided in detail in Table 9. The summary in the Discussion section refers to the results presented earlier.**

**Experimental design**

Research question is well defined, relevant and meaningful and within the scope of journal. Methods have been described in detail. However, readers would want to know more about the factors affecting motivation so as to see if anything can be done to enhance this. Detailed analysis with discussion of the factors need to be added so that research can help in improving the motivation levels in the long term.

**Response: We thank the reviewer for this positive statement. We elaborated on the discussion section. Please keep in mind that the questionnaire was newly developed and that this was the first approach to test an validate it. Further research is currently done to investigate the factors in more detail about what we can currently only speculate on.**

**Validity of the findings**

Conclusions of the article are well stated and supported by data, however, same is not true of the abstract conclusion which would need to be improved as stated above.

**Response: As stated above the abstract was revised.**